# Influenza Vaccine Utilization: A Comparison between Urban and Rural Counties in Florida

**DOI:** 10.3390/vaccines10050669

**Published:** 2022-04-23

**Authors:** Abdullah A. Alalwan

**Affiliations:** Department of Pharmacy Practice, College of Pharmacy, Qassim University, Buraydah 51452, Saudi Arabia; alalwan@qu.edu.sa

**Keywords:** urban, rural, influenza vaccine, vaccination utilization, Florida, underutilization, flu vaccine

## Abstract

**(1) Background:** The World Health Organization (WHO) and the Centers for Disease Control and Prevention (CDC) recommend that every person aged six months and over receive the influenza vaccine every year. Previous studies indicate that rural-area residents have less access to preventative health care services. This study aims to examine the variation in influenza vaccine use among rural and urban counties in Florida. **(2)** **Methods:** The study studied 24,116 participants from the Behavioral Risk Factor Surveillance System database. The study included only patients who live in Florida. We performed logistic regression analysis using survey procedures available in SAS^®^. Our regression model assessed the association between receiving the influenza vaccine and county status, age, income level, education level, and health coverage. We used ArcGIS software to create prevalence and vaccination maps. **(3) Results: **Of the total number of the study participants, 45.31% were residents of rural counties, and 54.69% were residents of urban counties. The logistic regression model showed no significant association between residing in rural counties and not receiving influenza vaccine in the past year (−0.05560, *p*-value = 0.0549). However, we found significant associations between not receiving influenza vaccine and age, high education level, and not having health care coverage (−0.0412, *p*-value < 0.0001; −0.04462, *p*-value = 0.0139; and 0.4956, *p*-value < 0.0001, respectively). **(4)**
**Conclusions:** Our study did not find an association between influenza vaccine use among rural and urban residence. Increasing age, higher education, and having health care insurance had positive associations with influenza vaccine use.

## 1. Introduction

Influenza is a viral infection that affects approximately 8% of the total adult population worldwide every year [1]. The average number of hospitalizations due to influenza is about 115,000 patients per year [2]. Furthermore, the number of annual deaths caused by influenza infection is about 500,000 persons worldwide [1]. The health and economic burdens of influenza are major concerns for the health care system. According to the Centers for Disease Control and Prevention (CDC) report released in 2011, an estimated 1532 deaths were caused by the influenza virus in the United States [3]. The annual influenza-related costs reach up to USD 167 billion, including treatment, hospitalizations, deaths, and lost works days [1]. These health and economic burdens urged the Advisory Committee on Immunization Practices (ACIP), which is administered by the CDC and the World Health Organization (WHO), to recommend those aged six months and older and high-risk patients to receive influenza vaccines every year [1,4]. Clinical guidelines for many diseases recommend the annual influenza vaccines to avoid the harmful consequences of infection. CDC has increased its efforts to establish vaccination campaigns for influenza to reach out to the public and promote the awareness of the importance of influenza vaccination. In addition, current changes in policy and regulations, such as enabling pharmacists to administer the influenza vaccines, have been implemented to increase access to influenza vaccines [5,6,7]. Applying such practices can help reduce the occurrence of influenza and consequently lessen the impact of the disease on public health and the economy. Despite these efforts, several recent studies reported underutilization of the influenza vaccine and suggested strategies and practices to promote vaccination, especially in high-risk populations [8]. This study aims to assess the geographic variations of the utilization of the influenza vaccine in Florida. The main goal is to compare the utilization of influenza vaccine in rural counties to that of urban counties across the state. This study will examine whether living in a rural or urban area has a significant impact on influenza vaccine utilization. Identifying the trend of the utilization of influenza vaccines can help us address unmet needs in areas with lower influenza vaccine use. Exposing the causes of utilization variation can help with implementing health care interventions to bridge the gaps in the areas with underutilization. The study could also help us glean more understanding of the disparities in health care in rural and urban areas.

It seems clear that rural residents have less access to the health care system due to distance or perhaps the paucity of health care facilities in these areas. Amburgh et al. discussed some of the challenges that face people who live in rural areas when they seek acute health care [2]. Amburgh et al.’s study stated that rural residents are more likely to have less access to preventative health care services due to the long distance they must travel to reach health care facilities [2]. In addition, well-trained clinicians and health care specialists are more likely to work in urban areas. This makes it even more difficult for those who live in rural areas to have frequent encounters with clinicians and health care providers. Furthermore, awareness campaigns usually take place in urban communities, as urban residents are often the targets of these campaigns [2]. Therefore, we hypothesize that rural residents are less likely to be vaccinated against seasonal influenza compared to urban residents.

## 2. Materials and Methods

The Behavioral Risk Factor Surveillance System (BRFSS) database was used to conduct this study. BRFSS is a program that gathers information by surveying participants nationwide via phone calls. BRFSS is directed by the CDC and focuses its efforts on collecting data on behavioral and health measures. BRFSS provides information about influenza use in the past 12 months. It also provides geographical information, such as state of residence and county for each participant. The maps used in this study were obtained from the U.S. Census Bureau website. The Florida charts website, administered by the Florida Department of Health, was used to collect county-level information, such as income and education levels.

The geographic examination used ArcGIS software to create vaccination and prevalence maps to show the variation in vaccination rates based on the statuses of the counties, income levels, and education levels. The status of each county was determined using the 2018 Florida statutes definition of rural areas, which states that rural areas are those with “a population density of less than 100 individuals per square mile.” [9]. Data acquired from the Florida charts were combined with the maps obtained from the U.S. census. BRFSS data from the 2010 cycle were used to conduct the statistical analysis using SAS^®^ (version 9.4 SAS Institute Inc., Cary, NC, USA).

We conducted a Chi-square test to compare the baseline characteristics for the participants from Florida (Table 1). A logistic regression model was designed to analyze the relationship between the outcome variable and the predictor variables [10]. The outcome variable in our regression model is “receiving and not receiving influenza vaccine in the past 12 months.” The predictor variables in our model are “status of the county the participant lives in”, “age of the participant”, “gender of the participant”, “income level”, “education level”, and “having or not having health care coverage”.

We utilized survey procedures in SAS to account for the weight and stratum variables that were calculated to adjust for the sampling for each participant. Our results were reported as coefficient estimates and 95% confidence intervals with their *p*-values.

## 3. Results

We included (*n* = 26,114), in this study (Table 1). Among those participants, 45.31% (*n* = 11,832) were residents of rural counties, and 54.69% (*n* = 14,282) were residents of urban counties (Figure 1). Approximately 41.60% of rural county residents versus 46.70% of urban county residents reported that they had received an influenza vaccine in the past year. The vaccination map shows a lower rate of vaccination in the rural counties in the northern part of Florida; however, some rural counties, especially in the middle parts of Florida, showed high rates of vaccination. These high rates were actually higher than the rates in some urban counties (Figure 2). Some rural counties in the northern part of Florida had lower income levels compared to urban counties. The rural counties with lower income showed lower rates of vaccination, which may indicate a relationship between low-income level and the underutilization of the influenza vaccine (Figure 3).

The results of the logistic regression model did not show a significant association between receiving the influenza vaccine and the status of the county (−0.056, *p*-value = 0.0549). Age, high education level, and having health care coverage were significantly associated with influenza vaccination. Age and high education level (college graduate or higher) were negatively associated with not receiving influenza vaccine (−0.0412, *p*-value < 0.0001) and (−0.4462, *p*-value = 0.0139), respectively. In other words, older participants were more likely to receive the influenza vaccine in the past year. Uneducated participants and those with low education levels (up to high school education level) did not show significant estimate coefficients, which indicates that there is no association between lower education levels and influenza vaccine use. However, high education has a significant association with influenza vaccine use. Conversely, patients who did not have health care coverage were more likely to have not received an influenza vaccine in the past year (0.4956, *p*-value < 0.0001). Participants’ income levels did not show significant estimate coefficients for either the low-income levels or the high-income levels (Table 2). The findings from the logistic regression model suggest that being a resident of a rural or an urban county does not have an impact on the utilization of influenza vaccine in Florida. However, other factors were significantly related to the utilization of influenza vaccine use, such as age, high education level, and health coverage. These factors were associated with the vaccination status in the state of Florida after adjusting for the county information. 

## 4. Discussion

This study intended to examine the association of influenza vaccine use and the county status in the state of Florida. The results indicate that there is no significant association between receiving the influenza vaccine and being in either a rural or urban county. These findings did not correspond with the hypothesis, which predicted that rural residents would have lower rates of influenza vaccination. Additionally, these results differ from previous studies’ findings that show lower rates of preventative health care services in rural areas compared to urban areas [11].

The significant impact of age on receiving influenza vaccine shown in the results of this study agrees with the CDC statistics, which state that older populations have higher rates of influenza vaccination compared to younger adults [12]. Moreover, high education level has a significant impact on influenza vaccination, which seems intuitive because those with higher education are more likely to embrace healthy lifestyles compared to those with lower education levels [13]. Having health care coverage provides easier access to the influenza vaccine, which has been also proven by the findings of this study.

The study findings can be used to address the unmet needs of populations lacking access to the influenza vaccine. The Florida Health Department can utilize the findings of this study to improve the efforts to increase awareness of the importance of influenza vaccines among children older than six months and in younger adults. Awareness campaigns at schools and universities that target young people may bridge the gap and improve the utilization of influenza vaccine in these populations. Community education is also essential, especially in low-education groups and those that have limited access to health care providers. Implementing community-based health education programs is crucial to educate people on the importance of annual influenza vaccination, which can then increase the vaccine utilization and reduce the spread of the influenza epidemic spread, consequently limiting the health and economic burden of this infection in Florida.

This study addresses a vital and concerning health topic that is encountered every year. We examined the factors that can influence the utilization of the influenza vaccine using a reliable database. After ruling out the effect of living in rural and urban areas, we were able to explore the other factors that had significant impacts on influenza vaccination and focus our efforts on outreach to the affected groups. The findings of this study are believed to have good external validity because they include a large number of participants (*n* = 26,114). The large number of participants seems to be representative of Florida population. One of this study’s strengths is that it used survey procedures in SAS^®^ that account for stratum and weight variables, which may improve the validity and the results of the study.

The main limitation of this study is that it is based on a cross-sectional database that does not follow participants over time and, therefore, it does not provide detailed information on their influenza vaccine use in previous years. Additionally, the data utilized for this study were obtained from telephone calls, which limited the interactions between interviewers and participants. Furthermore, the exposure to the influenza vaccine was self-reported, so there is no method to validate the data obtained from participants. Missing data was a major challenge in this study. Many participants had missing values for county of residence. This made it harder to categorize them as rural or urban residents. There are several definitions for what classifies a county as rural or urban. Some sources may consider certain county rural while another may consider it urban, which can lead to misclassification of participants. Additionally, the income level and health coverage can be categorized based on the employer county instead of the participant residence county, which may introduce misclassification bias.

Future research may use the study findings as a basis to explore the disparity in health care among urban and rural areas. Additionally, this study may help to accelerate efforts to improve awareness of influenza vaccines, health education and promote health literacy in Florida and worldwide. This study can also support future work and policies regarding other vaccines and factors associated with the uptake of such vaccines, such as COVID-19 and human papilloma virus vaccines. New breakthroughs in producing vaccines, such as the use of new biomaterials and the new technologies including nanomaterial vaccines, will play an important role in the next generation of vaccines [14]. Therefore, the public and professional awareness about the new vaccines is important, especially the seasonal influenza vaccines and other respiratory vaccines that require frequent induction of immunity targeting the new strains of viruses. Moreover, the effect of inadequate health care coverage can be examined from data obtained after the implementation of the Affordable Care Act (ACA), which aims to provide health insurance to the entire population of the United States. Future studies that assess the utilization of the influenza vaccine after the launch of the ACA may help indicate the success of the new federal policies. Additionally, using and expanding the existing infrastructure and recently implemented policies during the COVID-19 pandemic can help to address the underutilization of influenza vaccines, especially in the underserved areas [15,16].

## 5. Conclusions

The underutilization of the influenza vaccine is a concern for clinicians, health care providers, and the U.S. Department of Health. Identifying the factors that contribute to this underutilization is crucial to help address the shortfalls that have led to such a deficit. This study tried to examine the factors that may cause the variations in influenza vaccine utilization. The main goal of this study was to identify whether urbanization in Florida counties has an influence on influenza vaccination. The study did not show any variation in influenza vaccine use based on county status. However, other factors had significant association with influenza vaccination, such as age, education level, and health care coverage. Addressing these factors will assist efforts aiming to improve influenza vaccine use. Promoting influenza vaccination should be a matter of concern. Concentrating efforts to tackle factors that hinder people from accessing the influenza vaccine will save financial resources in the health care system and enhance the overall health and wellness of the Florida state population.

## Figures and Tables

**Figure 1 vaccines-10-00669-f001:**
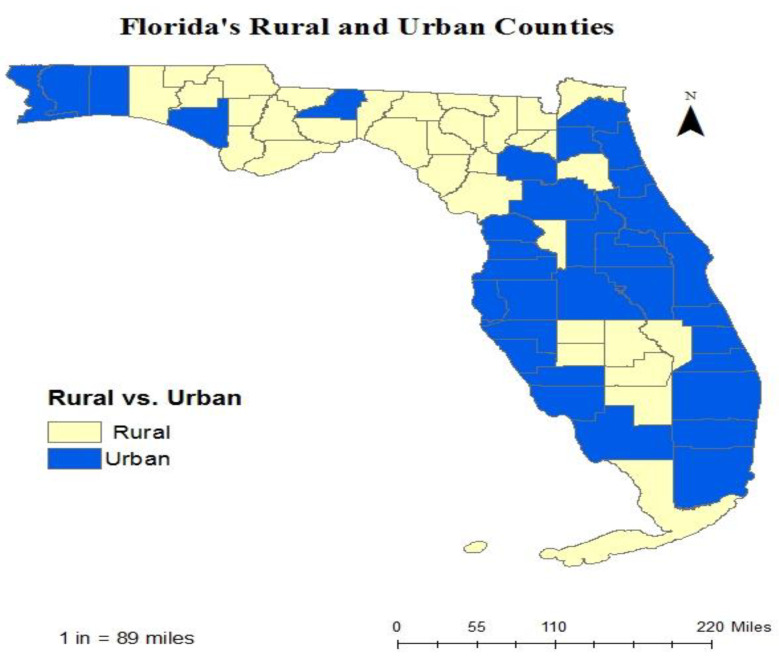
Urban and rural counties based on U.S. census.

**Figure 2 vaccines-10-00669-f002:**
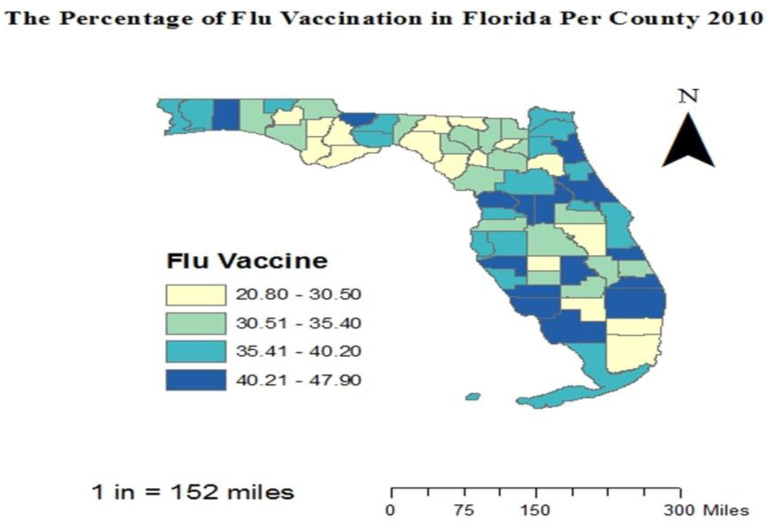
Vaccination rates per county in Florida.

**Figure 3 vaccines-10-00669-f003:**
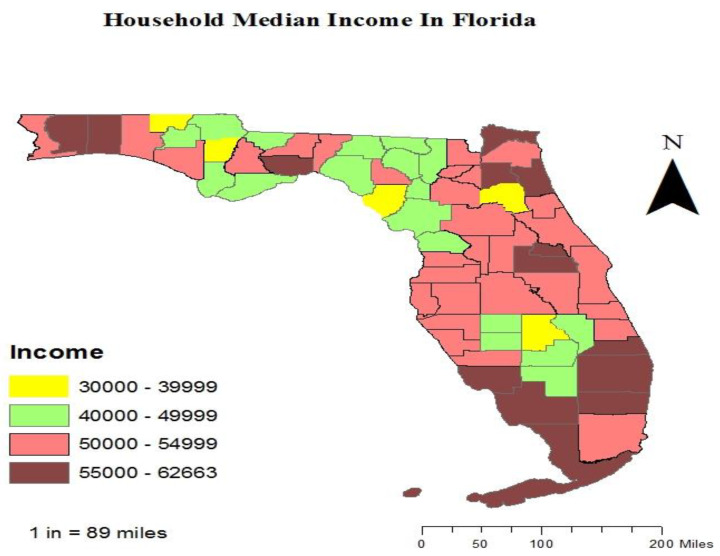
Median income map per county in Florida.

**Table 1 vaccines-10-00669-t001:** Participants’ baseline characteristics.

	Characteristics	Participants (%)	Flu Shot (Past Year) (%)		*p*-Value
			Yes	No	
Status	Rural	11,832 (45.31)	4922 (41.60)	6910 (58.40)	<0.0001
	Urban	14,282 (54.69)	6670 (46.70)	7612 (53.30)	<0.0001
Gender	Male	10,064 (38.54)	4507 (44.78)	5557 (55.22)	<0.0001
	Female	16,050 (61.46)	7085 (44.14)	8965 (55.86)	<0.0001
Education	Never attended school	16 (0.06)	4 (25)	12 (75)	<0.0001
	Grades 1 through 8 (Elementary)	639 (2.45)	266 (41.63)	373 (58.37)	<0.0001
	Grades 9 through 11	1733 (6.64)	621 (35.83)	1112 (64.17)	<0.0001
	High School Graduate	8123 (31.11)	3332 (41.02)	4791 (58.98)	<0.0001
	College 1 to 3 years (Some college)	7608 (29.13)	3367 (44.26)	4241 (55.74)	<0.0001
	College graduate or more	7995 (30.62)	4002 (50.06)	3993 (49.94)	<0.0001
Income	Less than USD 10,000	1750 (6.70)	527 (30.11)	1223 (69.89)	<0.0001
	USD 10,000 to less than USD 15,000	2036 (7.80)	853 (41.90)	1183 (58.10)	<0.0001
	USD 15,000 to less than USD 20,000	2412 (9.24)	1020 (42.29)	1392 (57.71)	<0.0001
	USD 20,000 to less than USD 25,000	2985 (11.43)	1333 (44.66)	1652 (55.34)	<0.0001
	USD 25,000 to less than USD 35,000	3305 (12.66)	1523 (46.08)	1782 (53.92)	<0.0001
	USD 35,000 to less than USD 50,000	4039 (15.47)	1891 (46.82)	2132 (53.92)	<0.0001
	USD 50,000 to less than USD 75,000	3947 (15.11)	1815 (45.98)	2132 (54.02)	<0.0001
	More than USD 75,000	5640 (21.60)	2630 (46.63)	3010 (53.37)	<0.0001
Coverage	Yes	22,596 (86.53)	10,997 (48.67)	11,599 (51.33)	<0.0001
	No	3518 (13.47)	595 (16.91)	2923 (83.09)	<0.0001
Total		26,114	11,592 (44.39)	14,522 (55.61)	<0.0001

**Table 2 vaccines-10-00669-t002:** Parameter estimates for not receiving influenza vaccine in the past year.

Parameter	Estimate	Standard Error	Pr < ChiSq
Intercept	3.3432	0.2244	<0.0001
Rural	−0.0560	0.0292	0.0549
Age	−0.0412	0.00254	<0.0001
Less than USD 10,000	0.1347	0.1357	0.3210
USD 10,000 to less than USD 15,000	−0.0251	0.1186	0.8323
USD 20,000 to less than USD 25,000	−0.0613	0.1065	0.5652
USD 25,000 to less than USD 35,000	0.0224	0.0991	0.8214
USD 35,000 to less than USD 50,000	−0.0987	0.1156	0.3933
USD 50,000 to less than USD 75,000	−0.0451	0.0759	0.5523
More than USD 75,000	0.0709	0.0782	0.3642
Grades 1 through 8 (elementary)	−0.04241	0.2955	0.1512
Grades 9 through 11 (some high school)	0.0279	0.2158	0.8971
High school graduate	−0.0733	0.1831	0.6890
College from 1 to 3 years (some college)	−0.2409	0.1828	0.1875
College graduate or more	−0.4462	0.1814	0.0139
No health coverage	0.4956	0.0626	<0.0001

## Data Availability

Not applicable.

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
