# Peer review of "Influenza Vaccine Utilization: A Comparison between Urban and Rural Counties in Florida"

_vaccines, 2022, doi:10.3390/vaccines10050669_

Round 1

Reviewer 1 Report

In this manuscript, the authors analyzed the association between influenza vaccination and county status, age, income level, education level and health coverage respectively among 24,116 participants in Florida. The results showed that influenza vaccination was not significantly associated with county status, but correlated with age, high education level and health coverage. In addition, the results showed a big variation in the percentage of influenza vaccinations received among different counties. These results have guiding significance for promotion of influenza vaccination. To make the results more convincing, the following adjustments is required:

  1. Due to the significantly different populations among different counties or districts, the authors should analyze the percentage of influenza vaccinations in rural and urban areas of each county, or re-analyze the overall data classified by cities and countrysides, so as to accurately reflect the vaccination between rural and urban areas.
  2. The results showed that vaccination was associated with age, high education level and health coverage. However, the authors did not make it clear whether the association was the same in each county.
  3. The results showed the association between univariate factors and influenza vaccination. The effects of multiple factors on influenza vaccination should be simultaneously analyzed. In addition, the results showed that some low-income rural counties had high vaccination rates (e.g. Highlands county), while high-income urban counties showed low vaccination rates (e.g. Broward county). The authors should analyze the reasons for this, which could be helpful in promoting vaccination.
  4. Authors should analyze the exact reasons by which it affects vaccination coverage rates, e,g, vaccination awareness, vaccination willingness etc.
  5. There are marking errors in the article, such as Figures 2 and 3 in the Results section on Page 3 were mistakenly marked as Figures 4 and 5. 
  6. The names of each county should be marked on the maps in Figures 1-3.

Reviewer 2 Report

1. It is suggested to add new research studies
2. what is the suggestion of this study for future works?
3. Please discuss the use of new biomaterials with new technologies including nanomaterials
4. Please add these references for your discussion part of the manuscript and bold your study novelty :
-Rashidzadeh, Hamid, et al. "Nanotechnology against the novel coronavirus (severe acute respiratory syndrome coronavirus 2): diagnosis, treatment, therapy and future perspectives." Nanomedicine 16.6 (2021): 497-516.

Round 2

Reviewer 1 Report

I understand the difficulties to resolve the questions I raised before. I agree with the authors revisions. It can be accepted as it is.